# Long-term trends in obesity and overweight in women in Ghana from 2003–2023
Laura Ann Gray [1,2] ✉, Joseph Prince Mensah [1], Magdalena Opazo Breton [3], Richmond Nii Okai Aryeetey [4], Isaac Boadu[4], Emmanuel Anongeba Anaba[4], Afua Atuobi-Yeboah [4] & Robert Akparibo [1]

## Abstract

**Background** In Ghana, overweight and obesity prevalence among women (20-49 years) reached 50% in 2022, increasing from 40% in 2014. This study aims to understand what has driven previous trends in overweight and obesity among women of reproductive age in Ghana and to predict future trends that can help inform policy making and public health surveillance.

**Methods** We used data from the Ghana Demographic and Health Survey (DHS) containing information on women of reproductive age (aged 15 to 49 years). Data collected between 2003 and 2022 provided cohorts born between 1953 and 2007. Age-period-cohort (APC) analysis was used to disentangle the effects of age, time, and generation on trends in the odds of obesity and overweight.

**Results** The prevalence of overweight and obesity increases during the study period, especially with age, in all cohorts. In the APC analysis, the odds of obesity increase with age until age 42–43 years (odds ratio (OR): 9.37; 95% confidence interval (CI): 5.63–15.59, compared to 20–21 year olds) before levelling out. Accounting for age and birth cohort, the odds of overweight and obesity increase significantly over time between 2003 and 2015 (overweight OR: 1.76; 95% CI: 1.47–2.11, obesity OR: 2.27; 95% CI: 1.77–2.91), after which the effect levels out and appears to stabilise. There is no effect of birth cohort on the odds of overweight or obesity.

**Conclusions** Although high, the increasing odds of obesity and overweight in Ghana appears to be stabilising. However, the increasing odds of obesity with age, suggest that an ageing population could mean that the prevalence will increase into the future.

## Plain Language Summary

Overweight and obesity are increasing rapidly in Ghana, especially among women, leading to serious health problems. We wanted to understand why these rates are rising and predict future trends to help create effective health programs. We looked at health survey data from Ghanaian women aged 15–49 collected over two decades (2003-2022). We used a specialist method to see how age, time, and the generation a woman was born into influenced her weight status. We found an increase in overweight and obesity among Ghanaian women over time, particularly between 2003 and 2014. Women tend to gain weight as they age, but we found that changes over time were more important than generational differences. Early interventions could be crucial to promote healthier weight and prevent long-term health issues for women.

Overweight and obesity rates are increasing rapidly in developing countries, creating an increased risk of non-communicable diseases. Globally, at least 43% of adults are overweight, and 1 in 8 people are living with obesity[1]. In Sub-Saharan Africa, overweight rates are increasing fastest among urban-dwelling adult women, especially those living in Southern Africa[1]. Our recent review and meta-analysis show that the pooled prevalence of overweight and obesity in Africa is 29.89% and 26.43% respectively in women, and 23.09% and 10.46% respectively in men[2]. In a similar review, a higher prevalence of a combined overweight and obesity (61.4%) was reported

among men and women with diabetes[3]. In Ghana, overweight prevalence among women aged 20–49 years has reached an all-time high of 50% in the latest National Demographic and Health Survey (DHS) in 2022[4]. Among men aged 20–49 years the rate was estimated at 21%. This rate is an increase from the 2014 estimate of 40% among women and 16% among men[5].

Overweight and obesity are linked with preventable deaths due to their association with increased risk of non-communicable diseases including cardiovascular diseases, Type II Diabetes, and some cancers[6]. The focus on women in this study is primarily due to their disproportionate burden of

[1]Division of Population Health, University of Sheffield, Sheffield, UK. [2]Healthy Lifespan Institute, University of Sheffield, Sheffield, UK. [3]School of Medicine, University of Nottingham, Nottingham, UK. [4]Department of Population Family and Reproductive Health, University of Ghana, Accra, Ghana. ✉e-mail: laura.gray@sheffield.ac.uk

these health conditions among women, coupled with the high prevalence of overweight and obesity within this demographic group. Existing literature has highlighted the urgent need for interventions and policies specifically aimed at reducing obesity in women of reproductive age in urban Africa[7]. Women and men experience different long-term health outcomes from obesity, particularly in areas such as reproductive health, cardiovascular health, and metabolic diseases. For instance, overweight and obesity in women are closely linked to complications like polycystic ovary syndrome (PCOS), infertility, gestational diabetes, and increased risks during pregnancy[8]. These gender-specific health concerns highlight the need for targeted research to understand how these conditions develop over time in women. Additionally, focusing on women enables us to better address the unique challenges and needs of this group, leading to more effective health interventions.

Non-communicable disease burden is also associated with reduced quality of life, and has an impact on mental health by exacerbating stigma, discrimination, limited social participation, and depression[9,10]. Management of overweight and obesity and their health outcomes demands a lot of financial[11] resources. Thus, increasing overweight and obesity poses a substantial challenge for health systems in low-income countries like Ghana where there is limited financial and intervention capacity to prevent and treat weight gain and the resulting morbidity and adverse societal outcomes.

Overweight is precipitated by diverse factors consistent with the nutrition transition. Ghana is already in an advanced stage of the nutrition transition[12], characterised by increased access to and consumption of unhealthy foods. Dietary patterns among Ghanaians, especially in urban dwellers are commonly influenced by easier access to fried foods, sugar-sweetened beverages, and highly processed foods[13–15]. Marketing of these foods to both adults and children and the steep price differential between unhealthy and healthy foods shifts the balance of decision towards consumption of unhealthy foods[16,17]. Further, increased access to motorised transportation and penetration of digital communication and other technologies has decreased physical activity and therefore energy expenditure[18].

The DHS of households in Ghana collects data on weight and height of adult women and men, enabling nationwide estimation of overweight and obesity. Given that overweight prevalence has been increasing rapidly following each of the DHS surveys since 2003, there is a need to determine what is driving the increases in overweight and obesity to help forecast future obesity and overweight prevalence and inform the development of interventions to tackle the issue.

Age-period-cohort (APC) models can be used to disentangle how age, time (period) and generation (cohort) influence trends in public health outcomes. APC models have previously been used to investigate trends in body mass index (BMI) or the odds of obesity and/or overweight in a number of settings; Australia[19], China[20], England[21,22], Estonia[23], France[24], India[25], Ireland[26], New Zealand[27], Taiwan[28] and the United States[29–31]. Previous literature has used APC to investigate obesity in women of childbearing age[25], but there is little evidence of what is driving these trends in African settings.

This study explores the long-term trends in obesity and overweight in women in Ghana. It aims to aid understanding of what drives these trends, help to determine what the trends of obesity and overweight are likely to be in the future in Ghana, and inform policymakers and public health surveillance[31]. The health system in Ghana is under pressure to address the rising rates of chronic diseases, many of which are linked to obesity[32,33]. Despite this, there is a substantial lack of population based research on historical obesity trends in Ghana, as well as other African populations. Women, particularly those of reproductive age, may experience health complications that can burden healthcare systems in unique ways. Researching the long-term effects of obesity on women can assist in strengthening healthcare strategies, ensuring that women receive the appropriate care and resources they need for prevention and management of obesity-related conditions.

We find that obesity and overweight are a growing problem in women of reprodctive age in Ghana, and that this is a much larger public health problem than underweight which has traditionally been more widely researched. We find no evidence to support interventions targeted at specific cohorts, but that a women's age has a significant influence on their odds of living with obesity. We find a significant increase in the odds of obesity as women get older, suggesting that interventions should be targeted early, before women reach childbearing age. The findings of this present study build upon our ongoing body of research, providing evidence of the financial implications of inaction in addressing overweight and obesity in Ghana. These findings are critical as we prepare to engage relevant stakeholders in discussions on how to more effectively address this issue in the country.

## Methods

This study utilised individual-level data from the Ghana Demographic and Health Surveys (GDHS), a nationally representative survey conducted by the Ghana Statistical Services with technical support from the DHS Program (Rockville, MD, USA)[34]. The DHS collects demographic data from over 85 low- and middle-income countries approximately every five years and provides essential information to monitor population health and demographics. The GDHS data are publicly available and can be accessed upon reasonable request through the DHS Program website (https://dhsprogram.com/data/). To access the dataset, we registered on the DHS Program website and submitted a brief description of our study, after which permission to download the anonymised data was granted. The GDHS survey protocols were approved by the Ghana Health Service Ethical Review Committee and the Institutional Review Board (IRB) of ICF International. Written informed consent was obtained from all participants during data collection. This study used secondary analysis of publicly available, anonymised data; therefore, no additional ethical approval or IRB approval was required for this analysis. The GDHS uses a stratified, two-stage cluster sampling design to ensure representativeness at both national and regional levels. Firstly, clusters are selected based on probability proportional to size, stratified by region and type of residence, followed by the selection of households within these clusters. Sampling weights are then applied to adjust for differences in selection probabilities.

The sample for this study comprised Ghanaian women of reproductive age (15 to 49 years old) whose BMI was measured during the DHS survey. Cross-sectional data were collected in 2003, 2008, 2014, and 2022, thus including participants born between 1953 and 2007[35–38]. BMI measurements were not adjusted for pregnancy and therefore pregnant women were removed from the analysis.

We used an age-period-cohort approach to studying long-term trends in obesity for women in Ghana. To deal with the 'identification problem' associated with APC models[39] related to the perfect collinearity[19] between age, period and birth cohort, we used a three-strategy approach based on previous literature[40]. First, we provide descriptive evidence on period trends and age trajectories by birth cohort. Second, we created grouped age and birth cohorts to use in our model, breaking the linear dependency between the variables. Third, we changed the groupings in the age and birth cohort variables to run a series of sensitivity analysis and check the robustness of our results.

The Ghana Statistical Service submitted the survey protocol to the Ethical Review Committee (ERC) of the Ghana Health Service to ensure that the survey procedures were in accordance with Ghana's ethical research standards. The ERC granted ethical clearance for the survey. The Informed Consent Form Institutional Review Board provided ethical clearance for the GDHS survey in accordance with U.S. and international ethical research standards.

Our outcome variables were obesity and overweight, which we defined using BMI thresholds. BMI was calculated using weight (kilogram) and height (metres), using the following formula:

$$BMI = \frac{weight\,(kg)}{height(m)^2} \qquad (1)$$

Participant weight was measured using SECA 874U scales with a digital display, and height was measured using a ShorrBoard® measuring

board. Adults in our sample were then classified as having obesity (BMI ≥ 30 kg/m²), overweight (25 kg/m² ≤ BMI < 30 kg/m²), a healthy BMI (18.5 kg/m² ≤ BMI < 25 kg/m²) or underweight (BMI < 18.5 kg/m²). BMI thresholds for children (under 18 years) differ by age and sex in accordance with the International Obesity Taskforce (IOTF) definitions for obesity and overweight[41] and underweight[42]. Following previous literature[22] we created two binary outcome variables for use in APC models (those classified as underweight were removed from our sample);
 (i) obesity vs healthy weight status and
 (ii) overweight vs healthy weight status.

Our APC approach determines our exposure variables: age, period and birth cohort. We created a grouped age variable with 2-year age groups ranging from 15–17 years until 48–49 years; a period variable consisting of the four years the survey was run (2003, 2008, 2014 and 2022); and a 10-year grouped birth cohort variable with the oldest birth cohort including those born between 1953 and 1959 and the most recent one, those born between 2000 and 2007.

### Statistics and reproducibility
A series of two-proportion z-tests were conducted to assess changes in obesity odds among the participants in the sample across the four survey years: 2003, 2008, 2014, and 2022. The test of equal proportions was used to compare the proportion of women with obesity between each pair of time periods.

Next, we described our sample and graphically explored the prevalence of obesity, overweight and healthy weight over time. We then graphically displayed the prevalence of obesity and overweight over age by grouped birth cohorts (10-year birth cohorts).

Finally, we estimated two APC models for our outcome variables (obesity and overweight), using a multivariate logistic regression for each binary outcome variable defined earlier (i and ii) in a similar way to previous analysis studying obesity using BMI[22], smoking behaviour[43] and alcohol abstention and consumption[44]. For each binary outcome, three covariates were included: age, period and birth cohort. These variables were grouped as described earlier in order to solve the 'identification problem'[22,43–45]. We use the earliest period (2003) and birth cohort (1953–1959) as the reference categories. For the age groups, we use 20–21 years as the reference because at this age, adult BMI is thought to be well-established[46,47].

We displayed the results from the logistic regression models graphically using odds ratios (ORs) displayed on a log scale with 95% confidence intervals (95% CI) to better visualise the changes in the odds of obesity and overweight by age, period and birth cohort. These figures show the trends in the odds of obesity and overweight driven by aging, time and generation, directly from results of the logistic regression model.

For sensitivity, we repeated our APC analysis using different groups for age and birth cohort (5-year age groups and 7-year birth cohorts) in order to ensure our results are robust to different groupings of our exposure measures. Results are reported in Figs. S1 and S2 in the Supplementary Information.

We carried out a supplementary APC analysis on underweight (BMI < 18.5 kg/m2) vs healthy BMI in order to give our main results some context. Results of this additional analysis can be found in Table S8 and Fig. S3 of the Supplementary Information.

Missing data is assumed to be missing at random after adjusting for sampling weights. All analyses were performed in Stata version 18 (StataCorp).

### Reporting summary
Further information on research design is available in the Nature Portfolio Reporting Summary linked to this article.

## Results
The analysis used nationally representative data from the Ghana DHS for the years 2003, 2008, 2014, and 2022, focusing on women aged 15–49 years. From the original samples consisting of 35,017 women, 2590 were excluded from the study due to being pregnant. After removing 11,590 observations due to missing or invalid BMI values, the final analytical sample included 20,837 women across the four survey years, of whom 1738 were classified with underweight (BMI < 18.5 kg/m²).

Table 1 shows the weighted summary statistics. In all years, the average age was 29–30 years, and the median birth year for each group increased with each survey year. The average BMI increased significantly from 2003 to 2014 (ANOVA; $F_{(3, 20833)}$ = 132.72, $p < 0.001$). Compared to 2003, mean BMI was significantly higher in 2008 (increase of 0.49 kg/m²), 2014 (increase of 1.74 kg/m²), and 2022 (increase of 1.95 kg/m²; all $p < 0.001$). Tukey-adjusted pairwise comparisons revealed no statistically significant difference between 2014 and 2022 (mean difference = 0.21 kg/m², $p = 0.376$).

The distribution of BMI categories, particularly obesity, changed significantly over time (chi-squared test, $p < 0.001$). The proportion of women with obesity increased from 8.01% in 2003 to 9.17% in 2008, although this change was not statistically significant (two-proportion z-test, $p = 0.071$). A significant increase was observed between 2008 and 2014, with obesity prevalence rising to 15.14% ($p < 0.001$). Although there was no statistically significant change in the average BMI between 2014 and 2022, the proportion of women with obesity still increased significantly to 18.47% ($p = 0.001$), indicating an upward trend in obesity prevalence over the period. We found little difference in our variables of interest across those with and without a valid BMI measure. Table S3 in the Supplementary Information shows the age and birth cohorts, regardless of whether they had a valid BMI, were very similar to those shown in Table 1.

### Period trends
Figure 1 shows the trends in the prevalence of each weight status, defined by BMI, over time. The prevalence of those with 'healthy' weight was decreasing over time from 65.55% (95% CI: 64.11–66.99%) in 2003 to 49.59% in 2022 (95% CI: 48.08–51.11%). Over the same period, the prevalence of obesity rose from 8.03% (95% CI: 7.19–8.83%) in 2003 to 9.17%

## Table 1 | Characteristics of Ghana Demographic and Health Survey participants in the study sample using weighted data

| | Year | 2003 | 2008 | 2014 | 2022 |
|---|---|---|---|---|---|
| Women | N | 4933 | 4455 | 4393 | 7056 |
| Age | Mean (SD) | 29.2 (9.76) | 29.1 (9.78) | 29.9 (9.77) | 29.8 (9.83) |
| Birth year | Median (IQR) | 1975 (1965–1982) | 1980 (1971–1987) | 1985 (1976–1992) | 1993 (1983–2000) |
| BMI (kg/m²) | Mean (SD) | 23.1 (4.57) | 23.6 (4.84) | 24.8 (5.31) | 25.0 (5.57) |
| Underweight (BMI < 18.5 kg/m²) | N (%) | 428 (8.70) | 349 (7.82) | 235 (5.41) | 467 (6.48) |
| Healthy weight (18.5 ≤ BMI < 25 kg/m²) | N (%) | 3250 (66.02) | 2782 (62.29) | 2365 (54.38) | 3628 (50.36) |
| Overweight (25 ≤ BMI < 30 kg/m²) | N (%) | 849 (17.24) | 926 (20.72) | 1090 (25.07) | 1775 (24.64) |
| Obesity (BMI ≥ 30 kg/m²) | N (%) | 395 (8.03) | 410 (9.17) | 659 (15.14) | 1333 (18.51) |

*N* number of observations, *SD* standard deviation, *IQR* interquartile range, *BMI* body mass index

**Fig. 1 | Period trends for proportion of women classified as having a healthy weight, overweight, obesity and underweight.** Study period 2003–2022. Analysis uses weighted data. Lines show prevalence of each weight status with 95% confidence intervals in shaded areas, *n* = 20837.

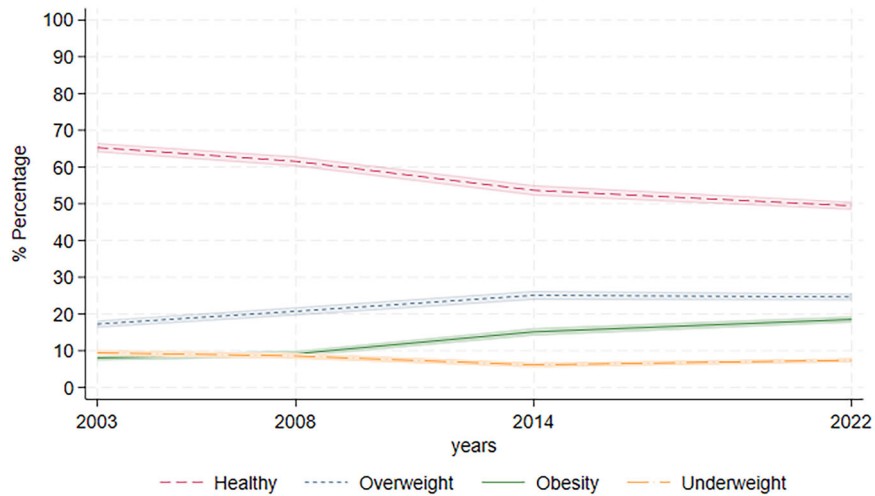

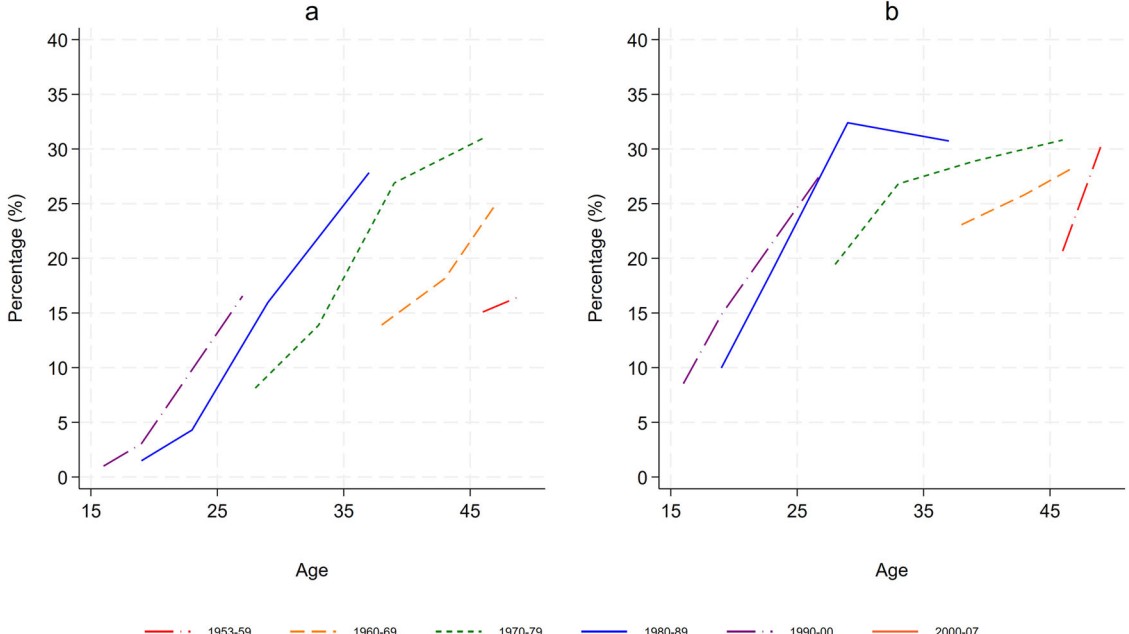

**Fig. 2 | Age trends by 10-year birth cohort for the prevalence of obesity and overweight. a** Prevalence of obesity and (**b**) prevalence of overweight, *n* = 20837.

(95% CI: 8.32–10.02%) in 2008 before a steep increase to 15.14% (95% CI: 14.08–16.20%) by 2014 and continuing the upward trend to (18.47%; 95% CI: 17.30–19.64%) in 2022. The prevalence of overweight also rose from 17.09% in (95% CI: 15.96–18.23%) in 2003 to 24.85% in 2014 (95% CI: 23.29–26.41%), and remained relatively stable to 2022 (24.57%; 95% CI: 23.27–25.87%). Conversely, the prevalence of underweight during this period decreased from 9.35% (95% CI: 8.47–10.23%) in 2003 to 6.10% (95% CI: 5.24–6.96%) in 2014, before increasing slightly to 7.37% (95% CI: 6.58–8.16%) by 2022.

## Age trends by cohort

Six 10-year birth cohorts were observed for between 1 (birth cohort group 2000–2007) and 19 years (1953–1959). Women born between 2000 and 2007 were only observed in the final survey in 2023 and were therefore only observed whilst under 23 years. Figure 2 shows the age trends in the prevalence of obesity and overweight for each 10-year birth cohort.

The prevalence of obesity (Fig. 2(i)) increased with age in all cohorts. There was a similar gradient in the increase in prevalence by age for each of the birth cohorts. Although the gradient was similar, there was a higher prevalence of obesity in younger birth cohort groups.

The prevalence of overweight (Fig. 2(ii)) also increased with age. This was steeper in younger women, less steep around the age of 30 years and became steeper again after 45 years of age. Similar to obesity, there was a higher prevalence of overweight in younger cohorts.

## Age, period and cohort effects

Figure 3 illustrates the age, period and cohort effects using the APC analysis for obesity and overweight, respectively. Full regression results can be found in Tables S1 (obesity) and S2 (overweight) in the Supplementary Information.

We found an increase in the odds of both overweight and obesity by age. The APC shows a higher increase in the odds of obesity compared to the increase in the odds of overweight. This also aligned with Fig. 2, which showed steeper lines by cohort for obesity than for overweight. Our analysis showed an increase in the odds of living with obesity from 15–17 years (OR: 0.34; 95% CI: 0.21–0.57, compared to 20–21 year olds), peaking at 38–39

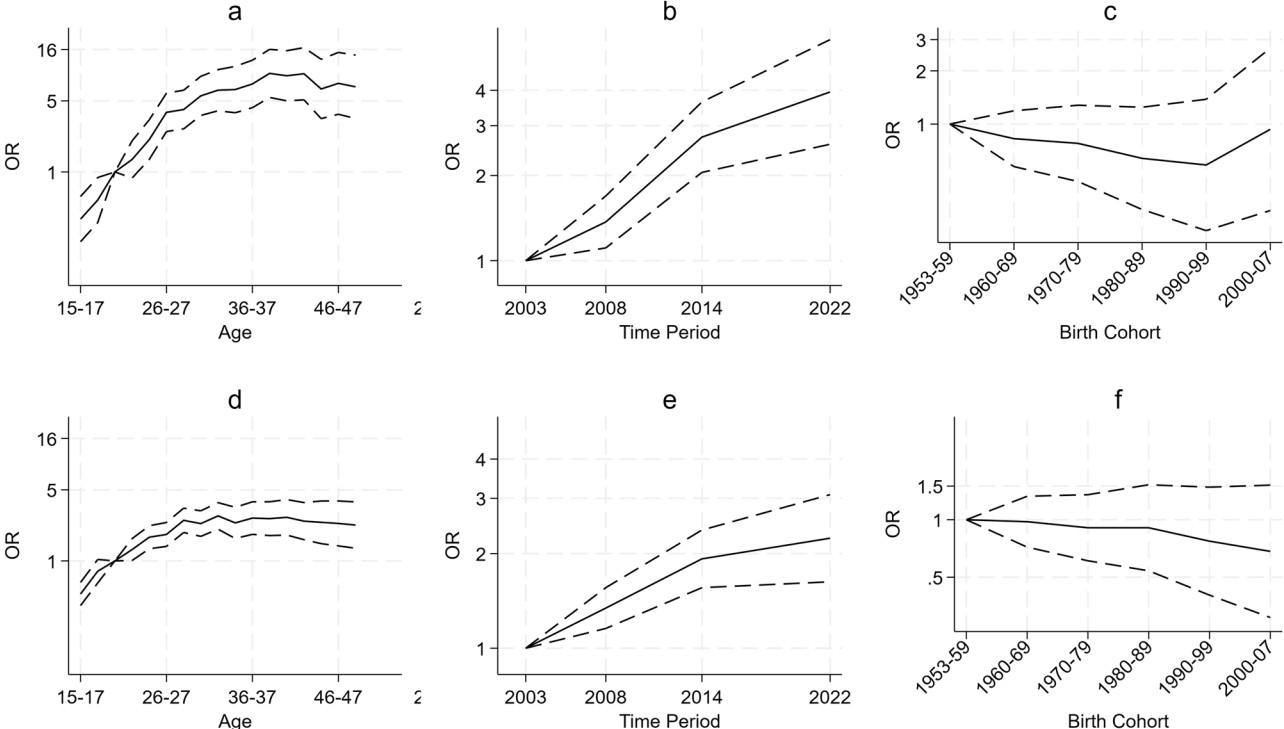

**Fig. 3 | Age, period and cohort effects.** Odds Ratios (OR) – solid line, 95% confidence intervals - dashed lines. **a** Age effects, **b** period effects, **c** cohort effects for the APC analysis for obesity (BMI > 30) vs Healthy weight (18.5 < BMI ≤ 25) (*n* = 14,998), **d** age effects, **e** period effects, **f** cohort effects for the APC analysis for overweight (25 < BMI ≤ 30) vs Healthy weight (18.5 < BMI ≤ 25) (*n* = 16,953).

years (OR: 9.32; 95% CI: 5.41–16.05), before decreasing slightly to the age of 48–49 years (OR 6.88; 95% CI: 3.36–14.10).

The increase was larger for the odds of obesity than for overweight. For overweight, the odds increased from 15–17 years (OR: 0.48; 95% CI: 0.39–0.60), peaking at 40–41 years (OR: 2.68; 95% CI: 1.79–4.01) and remained similar until 48–49 years (OR: 2.25; 95% CI: 1.33–3.80).

The period effects from the APC analyses were similar to those in Fig. 1; we found an increase in the odds of both obesity and overweight over time, which slowed after 2014. Again, the APC analysis for obesity showed a steeper increase over time in the odds of living with obesity than that of overweight. There was a relatively steep increase in the odds of obesity until 2014 (OR: 2.73; 95% CI: 2.05–3.64) compared to 2003 and a slower increase from 2014 to 2022 (OR: 3.95; 95% CI: 2.58–6.04). Similarly, there was a relatively steep increase in the odds of being overweight until 2014 (OR: 1.92; 95% CI: 1.56–2.38) and a slower increase from 2014 to 2022 (OR: 2.24; 95% CI: 1.63–3.08).

The odds of living with obesity was estimated to be at its lowest for those born between 1990-1999 (OR: 0.59; 95% CI: 0.25–1.38) and highest for those born between 2000–2007 (OR: 0.93; 95% CI: 0.32–2.69), both compared to 1953–1959, but there was no significant difference across birth cohorts. The odds of overweight also did not differ significantly between birth cohorts, with a low for those born between 1990–1999 (OR: 0.59; 95% CI: 0.25–1.38, compared to 1953–1959).

Although this study focused on obesity and overweight, Ghana also has a substantial prevalence of underweight in the population, resulting in a 'double burden of malnutrition'[48]. For this reason, it is important to view trends in obesity and overweight in a wider context. An APC analysis investigating trends in underweight (BMI < 18.5 kg/m$^2$) vs healthy BMI is illustrated in Figure S3 of the Supplementary Information with full regression results displayed in Table S8. There is a slight decrease in the odds of underweight with age, although this is not significant. Similarly, there is no significant change in the odds of underweight over time. There is, however, a significant difference in the odds of underweight by birth cohort,

with those born after 1970 having reduced odds compared to those born between 1953 and 1959.

### Sensitivity analyses
Results from APC analysis with different groupings showed similar results, suggesting that our results were robust to different groupings of our age, period and birth cohort variables. Results for the sensitivity analysis can be found in Tables S4-S7 and Figs. S1 and S2.

### Discussion
To the best of our knowledge, this study is the first to use an APC approach to disentangle the influence of age, time and generation on obesity and overweight trends in women of reproductive age in Ghana. It used nationally representative data from the Ghana Demographic and Health Surveys (GDHS) between 2003 and 2022. The high prevalence of obesity found in our sample from the GDHS aligns with trends observed across Sub-Saharan Africa, where the pooled prevalence of over-nutrition among women of reproductive age is 34.79%, comprising 21.81% with overweight and 12.99% with obesity[49]. Similarly, projections indicate that by 2030, nearly half of the women in Africa will be living with obesity or overweight, highlighting a substantial public health concern[49].

The prevalence of both obesity and overweight increased between 2003 and 2014 but remained stable afterwards. Both the odds of living with obesity and overweight were very similar across most birth cohort groups and only differed for those born between 2000–2007. Our results suggest there was a substantial decline in the prevalence of women with healthy weight between 2003 and 2014. This was mirrored by an increase in the prevalence of both obesity and overweight, suggesting that women were moving from a healthy weight to overweight at a higher rate than they were from overweight to obesity. This was in contrast to previous studies, from high-income countries, where overweight remained stable due to the rate at which people moved from healthy to overweight being similar to the rate at which people moved from overweight to obesity[22]. This could potentially be

due to changing perceptions about a healthy body weight in Ghana[50,51]. In Ghanaian society, cultural norms and values often associate larger body sizes with wealth, beauty, and social status, particularly among women. This cultural preference substantially influences women's attitudes toward maintaining a normal body weight. Studies have shown that Ghanaian women often prefer a body size slightly above normal, associating larger body types with positive attributes such as affluence and high social value[51,52]. This preference can influence women's perceptions of health and weight management, potentially obscuring underlying nutritional issues and the long-term health consequences associated with overweight and obesity. Our results showed the odds of women living with obesity increased faster with age compared to overweight, similar to findings from previous studies from high-income countries[19,22].

Our age and period effects for obesity were similar to those found among urban women in a previous study, in a similar low middle income country context, in India, investigating the odds of obesity in women of reproductive age[25]. However, the Indian study found some significant cohort effects, albeit with no particular pattern. To the authors' knowledge, there are no comparable studies using APC to investigate obesity or overweight prevalence in Ghana or any other African country. Our oldest cohort, born between 1953-1959 saw a decrease in the odds of obesity alongside an increase in the odds of overweight. This could be due to a decrease in total body mass as women get older, leading to weight loss in those who previously had obesity, or could simply be due to lower numbers of observations in this age group.

Our results showed a continued increase in the odds of obesity, and although this has slowed in recent years, it continued to increase throughout the study period. This was a similar pattern to that seen in other countries, for example England[19,20,22,25,31]. A recent study in sub-Saharan Africa analysing DHS data has also reported similar findings[49]. Although the patterns were similar, the odds of obesity and overweight were much lower overall than in many other studies conducted, especially in high-income countries. However, given that the present study looks only at women to the age of 49 years, results are comparable with other studies for this age group[19,22].

We chose to use a three-strategy approach to deal with the 'identification problem' inherent to APC models, by providing descriptive figures, categorised age, period and cohort variables, and running a series of sensitivity analyses. The use of categorical age, period and birth cohort has been extensively used in similar literature[22,43–45], but other approaches are available, such as the use of smoothing splines[53,54]. We chose to use the categorical representation of our age period and cohort variables over splines because categorical variables offer greater transparency, clearer interpretation and avoids the risk of overfitting or masking relevant non-linearities.

The study had data limitations. Height and weight data of women in the GDHS for BMI calculations only went back as far as 2003 and data from previous years would have allowed the investigation of a wider time period and cohorts. Similarly, there were only four periods of data collection so although there was a good range of time covered by the data, more time points would have allowed the changes over time to be investigated in more detail. The GDHS contains repeated cross sections, so it is not possible to investigate temporal changes in obesity and overweight at an individual level; however, the repeated cross-sectional data allow us to investigate temporal changes at a population level. Using repeated cross sections also means that our data does not suffer from attrition bias.

The use of odds ratios following logistic regressions can overestimate the strength of the changes in prevalence over time if interpreted as proxies for prevalence ratios[55], an approach taken by similar studies in the past[25]. Our APC results should be interpreted only as odds ratios and not as proxies for prevalence ratios or used to estimate risk ratios. Our analysis uses a study design which is longitudinal at the population level, but studies which are purely cross sectional should consider the use of prevalence ratios or risk ratios. We are particularly interested in the trends and patterns of obesity and overweight over time, rather than the specific coefficient estimate and it is in this analysis of trends and

compatibility with similar studies that our study contributes to the existing literature.

Additionally, there was missing data regarding the measurement of individuals' height and weight in each GDHS data, as some participants refused or were unavailable for these measurements. We did not explore methods to account for the missing data and assumed that data was missing at random. However, we believe the data still provides a fair representation of the population despite this limitation. Although we excluded pregnant women from the analysis, we did not exclude those who had recently given birth. This subgroup was a small proportion of the sample, so their inclusion is unlikely to have substantially impacted the main results. Although this study uses data which is generalisable to the Ghanaian population, including the use of sampling weights, further research is required to determine whether there is evidence for similar findings in Africa more generally.

In this study, we found less difference across birth cohort groups than across period and age groups, comparable with previous studies investigating similar generations[22]. However, previous studies have found cohort effects when looking at cohorts born earlier than those available in the GDHS[22]. This suggests that age and changes over time are more important than generational differences in Ghana over our study period.

## Conclusion

The reduced odds of underweight found in more recently born cohorts, coupled with the trend in prevalence of underweight in Fig. 1, which is decreasing and much lower than obesity or overweight, emphasises that obesity and overweight are of more concern to public health in Ghana, despite historically being less well-researched.

The lack of cohort effects for obesity and overweight found in this study suggests there is no evidence to support the need to target interventions at specific cohorts. There may however, be reasons to target women at certain ages, due to the significant age effects. The relatively steeper increase in age effects on obesity compared to overweight in younger women suggests that interventions should be targeted early, during childhood, before women reach childbearing age.

The increasing period effects between 2003 and 2014 and their stability between 2014 and 2022 suggest that future policy should consider population-based interventions to revert the curve by targeting the wider known determinants of obesity.

## Data availability

The data used in this study are publicly available and can be requested from the Demographic and Health Surveys (DHS) Program website (https://dhsprogram.com/data/). All figures and analysis in this manuscript have been created using the GDHS data which can be accessed as above. Source data for Fig. 3 is available in Supplementary Data 1.

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

## Acknowledgements

L.A.G. was funded by the Medical Research Council (MRC) [MR/S009868/1] and the Novo Nordisk Foundation [NNF24SA0090440]. M.O.B. was funded by the MRC [MR/T032472/2] and a University of Nottingham Research Fellowship. We would like to thank the University of Sheffield's Global Engagement Partnership Funding Scheme for their support. We thank the 'Measure DHS program' for providing free access to the Ghana Demographic and Health Survey (DHS) data. We also thank those who took part in the GDHS surveys for their participation. For the purpose of open access, the author has applied a CC BY public copyright licence to any Author Accepted Manuscript version arising from this submission.

## Author contributions

L.A.G., R.A., and R.N.O.A. conceptualised the study. L.A.G. and M.O.B. designed the study. L.A.G. carried out the analysis and drafted the article. J.M. and E.A.A. acquired the data and performed descriptive analysis of the data. M.O.B., R.A., R.N.O.A. and J.M. substantively revised the article. I.B., A.A.Y., R.N.O.A. and R.A. provided context specific to the Ghanaian population. All authors reviewed and approved the final version.

## Competing interests

The authors declare no competing interests.
