## [Transparent Peer Review file · Communications Medicine]

Long-term trends in obesity and overweight in women in Ghana from 2003–2023

Corresponding Author: Dr Laura Gray

Version 0:

Reviewer comments:

Reviewer #1

(Remarks to the Author)
Comments:

The manuscript by Gray et al. conducted a study to examine Long-term trends in obesity and overweight in women in Ghana (2003–2023) using DHS data. The authors found that Accounting for age 45 and birth cohort, the prevalence of overweight and obesity increased significantly over time between 46 2003 and 2015 (overweight OR: 1.76; 95% CI: 1.47-2.11, obesity OR: 2.27; 95% CI: 1.77-2.91). No effect of the birth cohort on the prevalence of overweight was observed. Overall, the manuscript is generally well-written, and the topic is of interest but not novel. However, I have a few minor comments that need to be addressed before the manuscript can be considered for publication.

Since this is a cross-sectional with a high prevalence of obesity, the odds ratios may have overestimated the strength of the association. It will be better to use the Prevalence Ratios instead of the Odds Ratios. The prevalence seems too high. Please double-check.

The authors didn't mention whether they have incorporated and conducted complex survey design analysis such proc surveylogistic regression to account for the complex survey design. Since this is a cross-sectional study, the temporal relationship cannot be addressed and needs to be mentioned in the manuscript.

The statistical analysis section needs a little more detail. Especially, for the restricted cubic spline models. What was the cutoff for your knots? There was no mention of RCS in the manuscript.

Reviewer #2

(Remarks to the Author)

The manuscript by Laura et.al investigated the long-term trends of obesity and overweight in women in Ghana. The manuscript is well written, and the methods are well chosen. However, I have some comments for consideration.

1. I appreciate that the authors applied Age-period-cohort (APC) analysis to examine the age, time, and generation in the prevalence of obesity and overweight. My concern is that the authors then used logistic regression models to examine the odds ratios of overweight or obesity. Logistic regression models work well with rare binary outcomes; I am concerned that the estimates of ORs might be biased given the prevalence of overweight and obesity in this population.

2. Why did the authors only investigate overweight/obesity rates among women in Ghana?

3. Line 147, the author mentioned "For the age groups we use 20-21 years as the reference case because at this age, adult BMI is thought to be well-established." Can the author provide a reference to support this argument?

4. Regarding the analytical sample, 1579 women were excluded due to underweight; however, I think it is important to include this group, at least in sensitivity analysis. It is also important to compare the sample with valid BMI values and those with missing values.

5. Results from sensitivity analysis should be provided.

6. Figure 2 is a bit difficult to read; the legends and colors for the cohorts are very similar. Please consider improving the figure in terms of readability. It is also interesting that there is a decrease of obesity for the cohort in 1960-69, but an increase of overweight. Have the authors thought about why?

7. Figure 1, please change the y-axis scale to 0-100, and add years in x-axis.

Minor comments:

Please add units after BMI across the manuscript.
Line 242-244: The sentence is too long; please consider rephrasing.

Version 1:

Reviewer comments:

Reviewer #1

(Remarks to the Author)

Thank you for addressing my comments. I have no further comments.

Reviewer #2

(Remarks to the Author)

The authors have addressed all of my previous comments thoroughly and have made appropriate revisions to the manuscript. I have no further questions or concerns. I recommend the manuscript for publication in its current form.

Response to Reviewers – Communications Medicine

We thank both reviewers for taking the time to review our article. We have addressed each point that the reviewers have raised and made appropriate changes to the article. Outlined below are our responses to each of the reviewers' points. We believe that the revised article is much stronger and we are grateful to both reviewers for helping us to improve the article.

Reviewer's Comment	Response from Authors
Reviewer 1	
The manuscript by Gray et al. conducted a study to examine Long-term trends in obesity and overweight in women in Ghana (2003–2023) using DHS data. The authors found that Accounting for age and birth cohort, the prevalence of overweight and obesity increased significantly over time between 2003 and 2015 (overweight OR: 1.76; 95% CI: 1.47-2.11, obesity OR: 2.27; 95% CI: 1.77-2.91. No effect of the birth cohort on the prevalence of overweight was observed. Overall, the manuscript is generally well-written, and the topic is of interest but not novel. However, I have a few minor comments that need to be addressed before the manuscript can be considered for publication.	We are pleased that the reviewer found the article to be well-written and of interest. We have addressed the reviewer's concerns regarding the novelty of the manuscript and have revised the background section to more clearly highlight the paper's novelty (see introduction, lines 72-82 and 110-122 and discussion, lines 305-315). We have outlined the specific needs for women of reproductive age¹ and the significance of this research within the context of Ghanaian and African research².
Since this is a cross-sectional with a high prevalence of obesity, the odds ratios may have overestimated the strength of the association. It will be better to use the Prevalence Ratios instead of the Odds Ratios. The prevalence seems too high. Please double-check.	We agree with you on your observation regarding the potential overestimation due to the exclusion of underweight women. We have now reanalysed the data with underweight women included in the sample. Women with underweight were excluded from this Figure 1 which meant the prevalence of obesity and overweight were over estimated. The updated findings are reflected in Figures 1 and 2 as well as the descriptive statistics in Table 1. In our revised analysis, the prevalence of obesity in 2022 in our data was 14.47%, which is consistent with data reported in the published literature³. Corresponding updates have been made to the and text (lines 214-223). Although ORs can over-estimate the strength of association when used to

calculate risk ratios or as proxies for PRs, in the context of common diseases⁴, we have not used ORs for this purpose in our analysis. We have revised our manuscript throughout to clearly communicate that our results are based on odds and should not be interpreted as proxies for PRs, an important clarification that was not adequately addressed in the original version. Furthermore, while our data consists of repeated cross-sectional surveys, we use a model that captures longitudinal data at the population level which is the primary level of interest in our analysis.

Our main reason for using ORs was to enable direct comparison of our results with existing literature. Several APC studies investigating obesity and overweight across various populations have employed ORs as the measure of association⁵⁻⁹. To maintain consistency and facilitate meaningful comparison, we choose to adopt same approach. Moreover, considering that BMI categories, reference group, and threshold values can be somewhat arbitrary, our focus is more on capturing overarching trends and patterns rather than emphasizing specific effect size.

We have incorporated these points into the discussion of our study's strengths and limitations. Additionally, we have discussed another article⁶ that used ORs to estimate risk ratios, noting the potential overestimation of association in that context. (lines 351-358)

Finally, to better illustrate our findings, and prevent over representation of the strength of association, we have changed the scale of the y-axes in Figure 3 (and equivalent sensitivity analysis). The y-axes now display the ORs on a logarithmic scale, preventing an exaggerated association which may

	have been the case in the previous version.
The authors didn't mention whether they have incorporated and conducted complex survey design analysis such as proc surveylogistic regression to account for the complex survey design.	Thank you for pointing this out. We have now updated the analysis to incorporate weights. We now account for the complex survey design employed by the GDHS by adjusting for sample weight, clustering and stratification (see lines 130-33).
Since this is a cross-sectional study, the temporal relationship cannot be addressed and needs to be mentioned in the manuscript.	Although the Ghana Demographic and Health Survey (GDHS) is cross-sectional, there are repeated cross sections at different time points which allow us to investigate population level obesity over time and the temporal relationship can be determined at a population level (which is what this study is interested in rather than individual changes). Using repeated cross sections also means that our data does not suffer from attrition bias. Additional text has been added to the discussion section to address this point (lines 347-350).
The statistical analysis section needs a little more detail. Especially, for the restricted cubic spline models. What was the cutoff for your knots? There was no mention of RCS in the manuscript.	We acknowledge the lack of clarity and apologise for this. We understand that the use of splines in APC analysis is sometimes used as a solution to the 'identification problem' in recent literature, however it does come with a loss of information¹ as does using categorical APC variables². We opted for a categorical representation of these variables because we were building on Opazo Breton and Gray³ who also grouped age, period and cohort to solve the identification problem. To be more precise about the approach used in our study, we have more explicitly stated our three-strategy approach in the Data and Research Design section (lines 141 – 144). We have also incorporated additional text and references to the Statistical Analysis section (lines 173-177), and added a discussion around these different methods, including relevant

	reference, to the discussion section of the article (lines 336-342).
Reviewer 2	
The manuscript by Laura et.al investigated the long-term trends of obesity and overweight in women in Ghana. The manuscript is well written, and the methods are well chosen. However, I have some comments for consideration.	We thank the reviewer for their comments and we are pleased that they feel the manuscript is well written and that the methods are well chosen. We hope that we have addressed your comments appropriately.
1.I appreciate that the authors applied Age-period-cohort (APC) analysis to examine the age, time, and generation in the prevalence of obesity and overweight. My concern is that the authors then used logistic regression models to examine the odds ratios of overweight or obesity. Logistic regression models work well with rare binary outcomes; I am concerned that the estimates of ORs might be biased given the prevalence of overweight and obesity in this population.	Please see our previous response to reviewer 1 which explains that we also considered using prevalence ratios but made the decision to use odds ratios to ensure international comparability of our results. ORs can indeed overestimate associations in common diseases if used as a proxy for prevalence ratios or risk ratios. We have made it clearer in our article that this is not our intention, and our results should be interpreted only as odds ratios and not as proxies for PRs or RRs. The ORs are not biased if interpreted as ORs rather than proxies for the prevalence ratio. Additionally, the main aim of this study was to compare trends in Ghana with studies from other populations. This would not have been possible if we were to use PRs because the majority of studies used ORs. (see discussion, lines 351-358) We have made changes to Figure 3, using a log scale for the ORs to prevent misinterpretation of our results.
2.Why did the authors only investigate overweight/obesity rates among women in Ghana?	We have expanded the background section with additional texts to explain why we focus on women in Ghana (lines 72-82). Obesity rates in Ghanaian women is a pronounced and growing public health problem but very under researched compared to underweight. Additionally, we have included a new ACP analysis of underweight in the supplementary material.

3.Line 147, the author mentioned "For the age groups we use 20-21 years as the reference case because at this age, adult BMI is thought to be well-established." Can the author provide a reference to support this argument?	Additional references^{10,11} have been added to the text as requested (line 177).
4.Regarding the analytical sample, 1579 women were excluded due to underweight; however, I think it is important to include this group, at least in sensitivity analysis. It is also important to compare the sample with valid BMI values and those with missing values.	Based on both reviewers' comments, we have now added underweight to our descriptive analysis. Table 1, Fig 1 and Fig 2 have been updated to reflect this. We thank both reviewers for bringing this to our attention. The reviewer raises an important point and we have now included analysis for underweight in the supplementary material (Table S8 and Figure S3). We chose not to include underweight in our main analysis because our primary focus was overweight and obesity - a condition that requires priority attention in Ghana. There is already a substantial amount of research into underweight in women in Ghana and we did not want to detract from the important findings in relation to obesity. In addition, the sample size for underweight was much smaller than for overweight or obesity. We have added a table with additional descriptive statistics (Table S3) to the supplementary material, which includes participants with missing BMI measures. The additional table shows little differences between the sample with missing BMI in Table 1 and the sample with all participants in Table S3. This has now also been discussed in the results section (lines 205-207).
5.Results from sensitivity analysis should be provided.	Full results from the sensitivity analysis can now be found in Tables S4-S7 as well as being illustrated visually in Figures S1 and S2 of the supplementary material.
6.Figure 2 is a bit difficult to read; the legends and colors for the cohorts are very similar. Please consider improving the figure in terms of readability.	We agree the choice of colour was inappropriate. We have changed the colours of the Figure so that the figure is more visually appealing.

It is also interesting that there is a decrease of obesity for the cohort in 1960-69, but an increase of overweight. Have the authors thought about why?	Our analysis shows an increase (rather than decrease) in both overweight and obesity prevalence for the 1960-69 cohort. We respectfully ask the reviewer to clarify this comment. Thank you.
7. Figure 1, please change the y-axis scale to 0-100, and add years in x-axis.	We have adjusted Figure 1 in line with the reviewer's comments.
Please add units after BMI across the manuscript.	Noted and amended throughout the article.
Line 242-244: The sentence is too long; please consider rephrasing.	This sentence has now been split into two sentences to ensure neither is too long. (now lines 289-292)

References

1. Pirotta S, Joham A, Grieger JA, et al. Obesity and the Risk of Infertility, Gestational Diabetes, and Type 2 Diabetes in Polycystic Ovary Syndrome. *Semin Reprod Med.* 2020;38(6):342-351. doi:10.1055/S-0041-1726866/ID/JR2000062-21/BIB
2. Amugsi DA, Dimbuene ZT, Mberu B, Muthuri S, Ezech AC. Prevalence and time trends in overweight and obesity among urban women: an analysis of demographic and health surveys data from 24 African countries, 1991–2014. *BMJ Open.* 2017;7(10):e017344. doi:10.1136/BMJOPEN-2017-017344
3. Tamir TT, Mekonen EG, Workneh BS, Techane MA, Terefe B, Zegeye AF. Overnutrition and Associated Factors among Women of Reproductive Age in Sub-Saharan Africa: A Hierarchical Analysis of 2019-2023 Standard Demographic and Health Survey Data. *Nutrition.* 2024;128:112563. doi:10.1016/j.nut.2024.112563
4. Gallis JA, Turner EL. Relative Measures of Association for Binary Outcomes: Challenges and Recommendations for the Global Health Researcher. *Ann Glob Heal.* 2019;85(1):137. doi:10.5334/AOGH.2581
5. Diouf I, Charles MA, Ducimetière P, Basdevant A, Eschwege E, Heude B. Evolution of obesity prevalence in France: An age-period-cohort analysis. *Epidemiology.* 2010;21(3):360-365. doi:10.1097/EDE.0b013e3181d5bff5
6. Chaurasiya D, Gupta A, Chauhan S, Patel R, Chaurasia V. Age, period and birth cohort effects on prevalence of obesity among reproductive-age women in India. *SSM - Popul Heal.* 2019;9(June):100507. doi:10.1016/j.ssmph.2019.100507
7. An R, Xiang X. Age–period–cohort analyses of obesity prevalence in US adults. *Public Health.* 2016;141:163-169. doi:10.1016/j.puhe.2016.09.021
8. Opazo Breton M, Gray LA. An age-period-cohort approach to studying long-term trends in obesity and overweight in England (1992 - 2019). *Obesity.* 2023;(31):823-831.
9. Reither EN, Hauser RM, Yang Y. Do birth cohorts matter? Age-period-cohort analyses of the obesity epidemic in the United States. *Soc Sci Med.* 2009;69(10):1439-1448. doi:10.1016/J.SOCSCIMED.2009.08.040
10. Cole TJ, Freeman J V., Preece MA. Body mass index reference curves for the UK, 1990. *Arch Dis Child.* 1995;73(1):25-29. doi:10.1136/ADC.73.1.25

11. De Onis M, Onyango AW, Borghi E, Siyam A, Nishida C, Siekmann J. Development of a WHO growth reference for school-aged children and adolescents. *الكامل لهذه املقالة*. العربية لهذه الخالصة يف نهاية النص *Bull World Health Organ*. 2007;85(9):660-667. . الرتجمة .
doi:10.2471/BLT.07.043497